# Comparing the Performance of Rolled Steel and 3D-Printed 316L Stainless Steel

**DOI:** 10.3390/mi15030353

**Published:** 2024-02-29

**Authors:** Yao-Tsung Lin, Ming-Yi Tsai, Shih-Yu Yen, Guan-Hua Lung, Jin-Ting Yei, Kuo-Jen Hsu, Kai-Jung Chen

**Affiliations:** 1Graduate Institute of Precision Manufacturing, National Chin-Yi University of Technology, No. 57, Section 2, Zhongshan Rd., Taiping District, Taichung 41170, Taiwan; train@ncut.edu.tw (Y.-T.L.); frankyen2003@gmail.com (S.-Y.Y.); 2Department of Mechanical Engineering, National Chin-Yi University of Technology, No. 57, Section 2, Zhongshan Rd., Taiping District, Taichung 41170, Taiwan; mytsai@ncut.edu.tw (M.-Y.T.); ray104250603@gmail.com (K.-J.H.); 3Department of Mechanical Engineering, Chien Hsin University of Science and Technology, No. 229, Jianxing Rd., Zhongli District, Taoyuan City 320678, Taiwan; l1060188@rechi.com (G.-H.L.); land1599512356@yahoo.com.tw (J.-T.Y.)

**Keywords:** additive manufacturing, subtractive manufacturing, metal 3D printing, surface roughness, rolled steel

## Abstract

Three-dimensional printing is a non-conventional additive manufacturing process. It is different from the conventional subtractive manufacturing process. It offers exceptional rapid prototyping capabilities and results that conventional subtractive manufacturing methods cannot attain, especially in applications involving curved or intricately shaped components. Despite its advantages, metal 3D printing will face porosity, warpage, and surface roughness issues. These issues will affect the future practical application of the parts indirectly, for example, by affecting the structural strength and the parts’ assembly capability. Therefore, this study compares the qualities of the warpage, weight, and surface roughness after milling and grinding processes for the same material (316L stainless steel) between rolled steel and 3D-printed steel. The experimental results show that 3D-printed parts are approximately 13% to 14% lighter than rolled steel. The surface roughness performance of 3D-printed steel is better than that of rolled steel for the same material after milling or grinding processing. The hardness of the 3D-printed steel is better than that of the rolled steel. This research verifies that 3D additive manufacturing can use surface processing to optimize surface performance and achieve the functions of lightness and hardness.

## 1. Introduction

Conventional computer numerically controlled (CNC) processes selectively remove material from a position to create desired geometric shapes in a process known as subtractive manufacturing. With subtractive manufacturing, it is challenging to make complex curves and micromachine parts. Therefore, a metal laser laminated manufacturing technology was created. That is additive manufacturing [1]. Additive manufacturing (AM) processes are innovative component manufacturing concepts for making complex curves and micromachine parts. AM processes can mainly be divided into two types. One is powder bed fusion (PBF), and the other is directed energy deposition (DED) [2,3]. Selective laser melting (SLM) is one of the powder bed fusion (PBF) technologies [4]. As PBF technology can produce customized internal flow channels or various products without forming molds or machining tools, it has been widely used and studied in recent years.

Metal parts have the advantage of resistance to high temperatures, high pressures, impact, and oil corrosion compared to conventional glass or plastic parts. Therefore, non-metal materials cannot replace them in some application situations. The surface morphology of PBF products is usually similar to that of porous materials, so they are not as good as subtractively manufactured products in performance metrics such as the surface finish, porosity, thermal deformation, and residual stress [5,6,7,8]. This phenomenon will affect the structure’s strength and the surface aesthetic issues. Furthermore, the workpiece’s surface performance will affect its distortion resistance, lubrication, air tightness, fatigue, and outward appearance [9]. So, controlling surface roughness is necessary when applying metal laser laminated manufacturing technology.

316L stainless steel is austenitic stainless steel. It is also called 316L SS. Due to its excellent mechanical properties and corrosion resistance, it is one of the most frequently used types of stainless steel. Many 316L SS components have complex geometries; examples include pipeline systems used in the nuclear industry, various tailor-made implants, automotive components, kitchen tools, and components used in the aerospace industry. They make conventional manufacturing processes difficult and costly. Zhang S. et al. [10] used 316L SS to replace A36 steel to repair a steel bridge beam by 3D printing. The experimental results showed that 316L SS can improve the tensile strength more than A36 steel. However, the 316L SS material has the disadvantage of lower yield strength. Zhang Y. et al. [11] used electron backscatter diffraction and transmission electron microscopy to analyze the deformation behavior of 316L stainless steel (316L SS) powder in a printed microstructure. The results showed that the printing parameters will affect the material behavior of 316L SS under PBF manufacturing. Wang Y.M. et al. [12] optimized printing parameters to change the density of a part. Moreover, Nath S.D. et al. [13] used laser powder bed fusion (L-PBF) to compare the mechanical properties of 420 stainless steel additives that were manufactured and subjected to heat treatment. The results showed that heat treatment can improve the tensile strength, yield strength, and elongation of 420 stainless steel parts, while the hardness cannot be changed. Based on the above description, the use of post-processing methods to improve the performance of metal 3D-printed specimens is a worthy issue to study.

This study compares surface roughness quality after grinding processes for the same material (316L stainless steel) between rolled steel and 3D-printed steel. The motivation for comparing surface roughness is that roughness is related to the performance of the processes of spraying, electroplating, and polishing [14,15]. Once the surface roughness becomes smooth, the performance and durability of the parts after surface treatment increase. This study chose milling and grinding because these methods are usually used in final surface processing. It is based on the 316L SS material and experiments with improving the surface finish on 3D additive manufacturing parts. In addition, we also explore the difference in the hardness, stress, and strain between rolled steel and 3D-printed steel. The purpose is to understand the properties of 3D-printed materials and applications.

## 2. Research Methods

The mechanical properties of SLM-fabricated alloys depend on the microstructure developed during processing [9]. To explore the mechanical behavior of SLM 316L SS and cold rolled 316L SS, this research used the substrate of the 316L SS to explore the difference in the performance of processing and properties between 3D additive manufacturing and subtractive manufacturing.

### 2.1. Laser 3D Printing Process

Laser 3D printing involves several parameters. Laser power, travel speed, and hatch spacing are three of the most easily manipulated. The hatch pattern, which affects the thermal stress profile, is easy to manage, but the effect is difficult to quantify. Layer-by-layer fabrication causes anisotropy, so building angle or component orientation becomes essential. Heat input is a function of several parameters. One definition of heat input is the energy density (*E*), as in Equation (1) [16].
(1)$$E=\frac{P}{\nu ht}$$
where *P* is the laser power in watts, *v* is the travel speed in mm/s, *h* is the hatch spacing in mm, and *t* is the layer thickness in mm.

### 2.2. Density and Porosity

Three-dimensional printing is a well-known technique that produces porosities in parts. It will reduce the density of the elements. To improve the density of 3D-printed parts, it is crucial to assess the physical origin of the different types of porosities and to measure the porosity rate as precisely as possible so that one may select the optimum manufacturing parameters. Porosity can be measured by Archimedes’ measurement [5]. The density of the fluid is *ρ_fluid_*, and that of the air is *ρ_air_* if we know the mass of the specimen in the air (*M_air_*) and in the liquid (*M_fluid_*). It becomes possible to calculate the density of the specimen. The formulas are Equations (2) and (3). (2)$$\rho_{specimen}=\frac{M_{air}}{\left(M_{air}-M_{fluid}\right)}\times \left(\rho_{fluid}-\rho_{air}\right)+\rho_{air}$$
(3)$$Porosity=1-(\frac{\rho_{measured}}{\rho_{theoretical}})$$
where *ρ_measured_* represents the measured density and *ρ_theoretical_* represents the theoretical density.

### 2.3. Experimental Materials

The study used the powder of Tongtai Corp. SS-316L to carry out the 3D printing of the metal materials. The manufacturer number is 3354574. The metal particle size is shown in Figure 1. The highest particle size distribution is about 36.5 ± 15.5 µm. Figure 2a is the schematic diagram of the metal 3D printing. The metal 3D printing facilities used Tongtai AMP-160 to print tensile and block specimens. Table 1 shows the printing processing parameters [17]. Figure 2b is the schematic diagram of subtractive manufacturing in the SS-316L plate. The original material dimensions of the cold rolled steel are W350 mm × L1219 mm × t3.0 mm and W1219 × L3048 mm × t10.0 mm. The tensile specimen’s original dimensions are W350 mm × L1219 mm × t3.0 mm. The impact specimen’s original dimensions are W1219 × L3048 mm × t10.0 mm. The cross-section reduction is about 15–25% after the rolling process. The grain size is about 20–50 µm. Figure 2c represents the standard scale of impact and tensile specimens. We used the subtractive process to manufacture the tensile and impact specimens according to the scale of Figure 2c. The scales of the impact and tensile specimens are according to the ISO 6892-1 design specifications [18].

Three-dimensional additive and subtractive manufacturing will produce warpage and distortion due to the hot and cold variation in processing [19]. As a result, 3D additive manufacturing completes parts by stacking them layer by layer. The materials will drop when printing in the hanging place. Therefore, support must be designed in the hanging area [20]. We use tensile specimens to explore the difference in the warpage and distortion when 3D metal printing is in support and non-support designs. The thickness of the tensile specimen is 3.0 mm. Figure 3a,b show the 3D metal printing specimens with support and non-support designs.

The research also discusses the secondary processing performance of rolled and 3D-printed steel in the 316L SS materials. The experimental specimens are used for the impact specimens to explore the difference in the weight, roughness, and hardness, as shown in Figure 4a,b. The Mitsubishi ML3015 eX-F Plus CO_2_ Laser facilities conducted subtractive manufacturing of the impact and tensile specimens. Rolled steel is manufactured by Ycinox Corp. Table 2 shows the chemical compositions of the rolled steel and 3D-printed powder steel made of 316L stainless steel.

### 2.4. Surface Roughness

Surface roughness is due to the high-frequency vibration factor producing an irregular surface in the processing [21]. After manufacturing the workpiece, we must measure the surface roughness to confirm the qualities. The research is according to the standard of ISO 25178 to measure the surface roughness of the workpiece [22]. Surface roughness is divided into arithmetic mean deviation Ra and ten-point average roughness Rz.

The middle arithmetic deflection of elaborated profile on the basic length Ra. Arithmetic mean deviation can be obtained by taking a standard-length l from the average line direction of the roughness graph. The X-axis is the middle line direction, while the y-axis is the roughness value. Ra is defined as Formula (4) when the roughness graph is *y* = *f*(*x*).
(4)$$Ra=\int_{0}^{l}f\left(x\right)dx$$

Ten-point average roughness Rz. The ten-point average roughness is obtained by taking a standard-length *l* from the average line direction of the roughness graph. The longitudinal direction is the expression of the roughness value. The sum of the averages can be obtained by adding the absolute mean value of the highest peak to the 5th peak *y_p_* and the absolute mean value of the lowest to the 5th bottom *y_v_*. Rz is defined as:(5)$$Rz=\frac{\left|y_{p1}+y_{p2}+y_{p3}+y_{p4}+y_{p5}\right|+\left|y_{\nu 1}+y_{\nu 2}+y_{\nu 3}+y_{\nu 4}+y_{\nu 5}\right|}{5}$$

In addition, the measurement of the surface roughness in the metal has been widely presented with 3D geometric shapes in recent years [9]. Therefore, the workpiece’s surface morphology can be given as *S_a_*, *S_q_*, and *S_z_*. *S_a_* represents the absolute mean value relative to the height difference at each point of the measurement surface. The formula is given as (6).
(6)$$S_{a}=\frac{1}{A}\iint \left|Z(x,\;y)\right|dxdy$$

*S_q_* represents the average value of the root mean square in each point height within the measurement range. The formula is given as (7).
(7)$$S_{q}=\sqrt{\frac{1}{A}\iint \iint_{A}Z^{2}\left(x,\;y\right)dxdy}$$

*S_z_* represents the sum of the maximum height peak and the total depth trough within the measurement range. The formula is given as (8).
(8)$$S_{z}={max}_{A}Z\left(x,\;y\right)+{min}_{A}Z(x,\;y)$$

## 3. Results and Discussion

### 3.1. Warpage and Distortion Analysis

Support scaffolding is used to aid support when the materials are under the floating state in additive printing. For example, the process involves a printing shell or surface [23,24]. In addition, the workpiece will produce warpage and distortion after the selective laser melting processing [2]. Therefore, we used simulation and experimental software to investigate the phenomenon of warpage and distortion when the specimen additive was manufactured with support and non-support scaffolding. The measurement facilities use KEYENCE VR-6000 3D optical profilometer to measure the warping and distortion in the tensile specimen after the tensile specimen of the 316L SS is manufactured by 3D additive and subtract processing. Figure 5 shows the simulation results of the tensile specimen 3D additive manufacturing process. Figure 5a was designed with support scaffolding, and the height of the support scaffolding was designed to be 3.0 mm. Figure 5b was designed without support scaffolding. The software for the simulation is the Ansys workbench 2022 R2. The boundary conditions of the simulation are set according to Table 1. We found that the support scaffolding design of the specimen is higher than the design without support scaffolding in the index of the warpage and distortion, as shown in Figure 5a,b.

Figure 6a,b show the tensile specimen with import support scaffolding and non-support scaffolding design after the 316L SS printing of the tensile specimen materials. Figure 6c shows the manufactured tensile specimens by laser subtraction. The average value of the warping and distortion is 0.46 ± 0.05 mm with support design, and the average value of the warping and distortion is 0.1 ± 0.03 mm without support design after the tensile specimen is manufactured by 3D additive printing, as shown in Figure 6a,b. The average value of the warping and distortion was 0.23 ± 0.03 mm when the tensile specimens were manufactured by laser subtract processing, as shown in Figure 6c. We realize that the support scaffolding will cause warping and distortion in the specimens when 3D additive processing is used. This is due to the uniform temperature distribution, leading to the warping and distortion in the specimens. It is also called the stress residual in the specimens [25,26,27].

### 3.2. The Porosity Analysis

The metal 3D-printed process will face the issue of porosity due to the powder thickness needing to be uniform [28]. It has been found that the porosities can be optimized by adjusting laser scanning paths and solid solution treatments to control the porosity size. Furthermore, the porosity will affect the workpiece’s weight and the reliability of structural strength [29,30]. The research manufactured tensile and block specimens to explore the weight difference between rolled and 3D-printed steel of the same 316L SS materials. The printing parameters are shown in Table 2.

Figure 7 shows the experimental results. Figure 7a shows the weights and measures instrument in the loading state. Figure 7b shows the weights and measures instrument in the loading state. The digital precision weights and measures instrument is manufactured by Nanxing Corp. Figure 7c shows the average weight of the tensile specimens. The blue bar is rolled steel, weighing 27.65 ± 0.1 g. The orange bar is 3D additive steel and weighs 24.04 ± 0.3 g. We found that the weight of the 3D additive manufactured material is lighter by 13.1% compared to the rolled material when the workpieces were manufactured into tensile specimens. Figure 7d shows the average weight of the block specimens. Rolled steel weighs 33.36 ± 0.2 g, as shown in the blue bar. The weight of the 3D additive manufactured material is 28.75 ± 0.3 g, as shown in the orange bar. The 3D additive manufactured steel specimens are lighter, by 13.83%, compared to the general rolled steel. These results show that the qualities of the 3D additive manufacturing are nearly the same in terms of porosity when the printing process is under the same laser power and scanner speed [28].

### 3.3. The surface Roughness and Hardness Analysis

The performance of the metal surface roughness will affect the morphology and assembly capabilities of the parts. The surface hardness of metal materials will affect the plastic deformation ability of the part. The surface roughness of the 3D additive manufactured specimen is over 13 μm in Ra and 88 μm in Rz. The surface roughness is higher than the rolled steel from the morphology observed in Figure 4. Ding H. et al. [31] and Natali S. et al. [32] represent hybrid manufacturing to improve the efficiency and surface quality of the 3D additive manufactured specimens. Therefore, we used the milling process to process the surface of the additive manufactured specimen and the rolled steel. Table 3 illustrates the conditions of milling processing. The purpose is to compare the performance in the surface roughness. The surface roughness index was measured by white light interferometry (ZYGO NewView8000). The material of the specimen is 316L SS. Figure 8a,b show the surface roughness in Ra and Rz when the block specimens were manufactured by 3D printing and rolling, followed by processing of the surface of the block specimens by milling machine finishing. The surface roughness of the 3D-printed steel is 0.592 μm and 2.941 μm in Ra and Rz. The surface roughness of the rolled steel is 1.269 μm and 4.289 μm in Ra and Rz. The experimental results show that the workpiece of 3D printing can achieve a fine metal composed of 316L SS after secondary processing.

We use the grinding process to optimize the surface roughness in 3D additive specimens to verify that it can also be used in another machining process to improve the surface roughness performance. The diamond grinding wheel is CBN325N100B, and the grinding process method is shown in Table 3. Figure 9 shows the experimental results. We observed that the surface roughness of the rolled steel is 0.569 μm and 3.104 μm in Ra and Rz after grinding of the surface of the specimens. The surface roughness of the 3D-printed steel is 0.202 μm and 1.283 μm in Ra and Rz after grinding of the surface of the specimens. The experimental results showed that the 3D-printed steel’s surface roughness is higher. However, they can use different machining processes to optimize the surface roughness of the parts.

The ratio of roughness depth Rz to the average surface roughness Ra represents the surface roughness performance. A high index in Rz/Ra is not suitable for application in aerospace and automobile parts [33]. Table 4 explains the performance of rolled and 3D-printed steel in Rz/Ra. We observed that the surface roughness performance is better when 3D additive manufactured specimens are milled or ground on the surface. However, the Rz/Ra index was higher than that of rolled steel. The 3D additive manufacturing process would produce pores in the specimens. Therefore, improving the porosity of the part is a critical technology in 3D additive processing.

In order to explore the reason why the performance of 3D-printed steel is better than that of rolled steel in terms of surface roughness after milling or grinding of the surface of the specimens, we used the FM-810e Microhardness tester to measure the hardness in the surface of the specimens, as shown in Figure 10a. The measurement specimens were rolled steel and 3D-printed steel, as shown in Figure 4. Figure 10b shows the measurement results of the 3D-printed steel. The hardness of HRC is about 35.3 ± 3.0. Figure 10c shows the measurement results of the rolled steel. The hardness of HRC is about 28.3 + 3.0/−1.5. In addition, we also used a metal tensile tester to test the performance of 3D-printed steel and rolled steel made of 316L SS. The metal tensile tester was manufactured by Chun Yen Corp., as shown in Figure 10d. Figure 10e shows the experimental results of 3D-printed and rolled steel in terms of the tensile strength and strain. We found that 3D printed steel is better than rolled steel in terms of the hardness and tensile strength index. However, the elongation materials are only below 25% compared to the rolled steel from the experimental results. This phenomenon is nearly the same as that of Basavraj, B. et al. [6]. It shows that the 3D-printed steel is a hard and brittle material, so post-processing can be used to optimize the surface roughness and fineness of the parts.

### 3.4. The Performance of the Morphology

Figure 11 and Figure 12 show the surface morphology when the specimens of 3D additive manufactured and rolled steel were milled and ground on the surface. We observed that the index of 3D additive steel is higher than that of rolled steel in Sz. It is due to the pore phenomenon during 3D additive printing [34]. The pore will increase the Sz index and reduce the specimen’s weight. It may cause fatigue and cracks in the parts when the parts are used in the dynamic [35].

## 4. Conclusions

Metal 3D printing technology has the advantages of faster prototypes, small batch production, no machining tools required, and significant material cutting loss. Therefore, is has been researched, developed, and applied extensively. However, this technology has some porosity, surface roughness, and other performance issues. This study uses the milling and grinding process to improve the performance of the surface roughness of the workpiece. From the experimental results, we obtained some conclusions, as follows:(1)The workpiece will produce warpage and distortion in the metal 3D printing process. This is due to the uneven cooling when the metal 3D printing process is under high-temperature sintering.(2)The 3D-printed workpiece is lighter by 13.5 ± 0.5% than the rolled steel when made of of 316L SS under the normal manufacturing process.(3)The porosity of the workpiece will increase the index of Sz in surface roughness. This phenomenon will affect performance and surface morphology.(4)The performance of 3D-printed steel is better than that of rolled steel in terms of tensile strength.(5)The hardness of 3D-printed workpieces is higher by 25% than that of rolled steel, and the tensile strength is higher by 34%. However, the ductility and malleability of 3D-printed workpieces are only 21% compared to the rolled steel made of 316L SS. Therefore, we found that a metal 3D-printed workpiece is a complex and brittle material compared to rolled steel [6].

We will study the following issues to explore the metal 3D printing process in the future:(1)Using different laser power and scanning speeds to improve the workpiece’s porosity and strength.(2)Using heat treatment to explore the microstructure variation and the performance in terms of the wear resistance.(3)Using the electroplating process to explore the ability and wear resistance of the electroplated layer to adhere to the surface of the 3D-printed workpiece.

## Figures and Tables

**Figure 1 micromachines-15-00353-f001:**
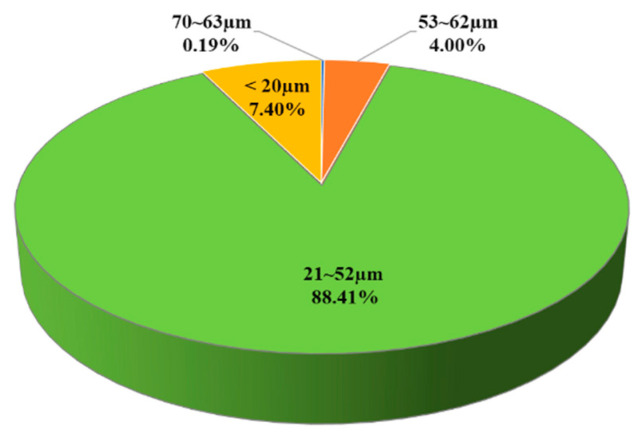
The particle size distribution percentage with the powder of metal 3D printing.

**Figure 2 micromachines-15-00353-f002:**
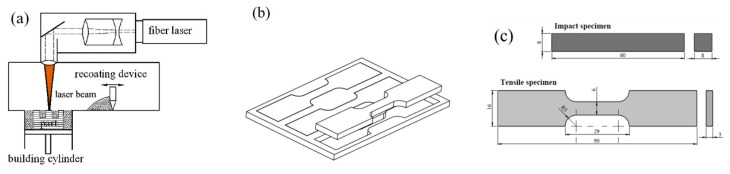
Additive and subtractive manufacturing with 316L stainless steel. (**a**). Additive manufacturing was used for the PBF process. (**b**). Subtractive manufacturing was used for the CO_2_ laser cut. (**c**). Specimen scale.

**Figure 3 micromachines-15-00353-f003:**
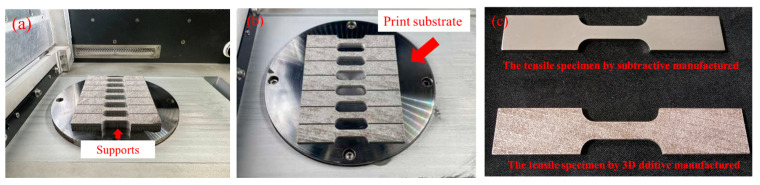
The 3D-printed metal material of 316L stainless steel (**a**). The 3D-printed tensile specimen with supports. (**b**). The 3D printing setup without support for the tensile specimen. (**c**). Tensile specimen.

**Figure 4 micromachines-15-00353-f004:**
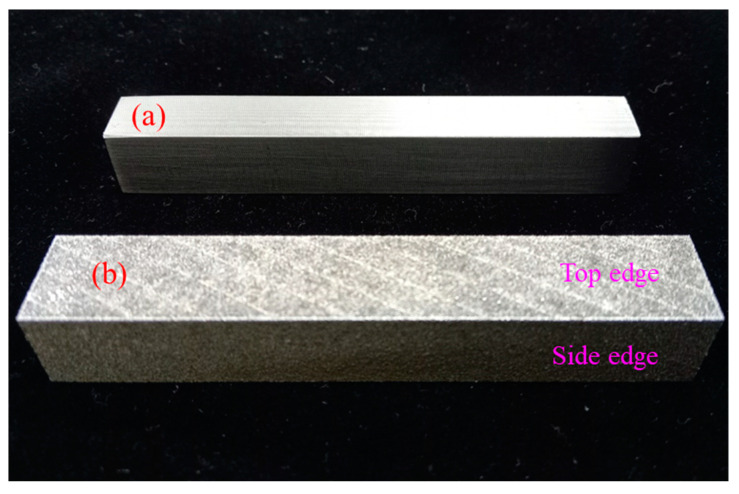
The impact specimens made of 316L stainless steel. (**a**). Cold rolled steel; (**b**). 3D-printed steel (3D additive manufactured steel).

**Figure 5 micromachines-15-00353-f005:**
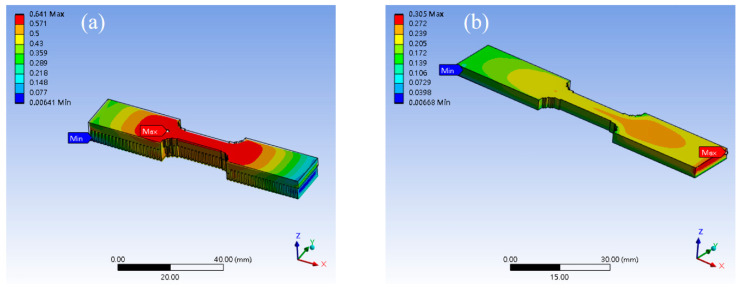
The simulation of 3D additive manufacturing specimen with support and non-support scaffolding design. (**a**) is the support scaffolding design. (**b**) is the non-support scaffolding design.

**Figure 6 micromachines-15-00353-f006:**
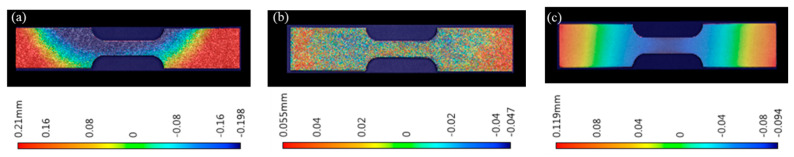
The warpage and distortion in the specimen of 316L staimless steel. (**a**). Tensile specimen by 3D additive manufacturing with supports. (**b**). Tensile specimen by 3D additive manufacturing without supports. (**c**). Tensile specimen by subtractive manufacturing.

**Figure 7 micromachines-15-00353-f007:**
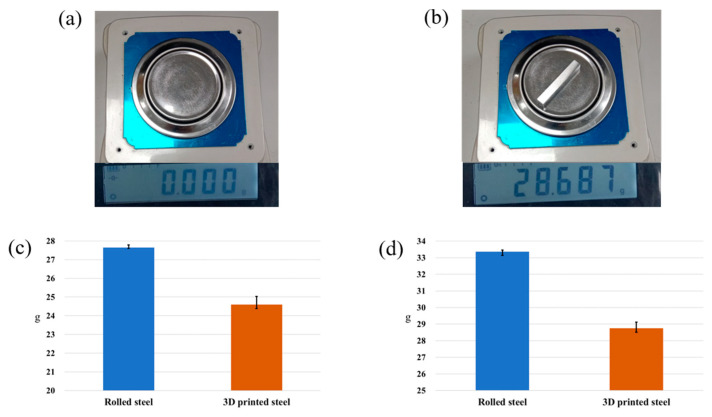
Digital precision weights and measures instruments measure the weight. (**a**) represents no loading in the weights and measures instrument. (**b**) represents the weight with the loading block specimen in the weights and measures instrument. (**c**) represents the weight of the tensile specimens. (**d**) represents the weight of the block specimens.

**Figure 8 micromachines-15-00353-f008:**
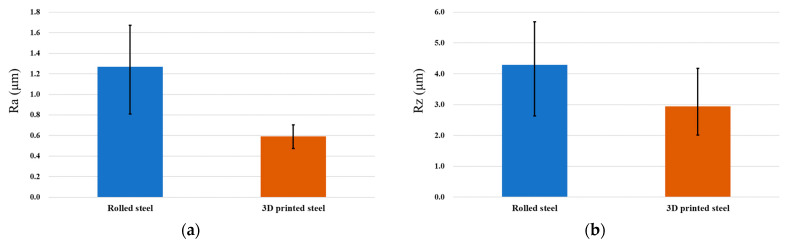
The surface roughness in Ra and Rz after the milling machine cutting of the surface of the block specimens. (**a**) The surface roughness in Ra. (**b**) The surface roughness in Rz.

**Figure 9 micromachines-15-00353-f009:**
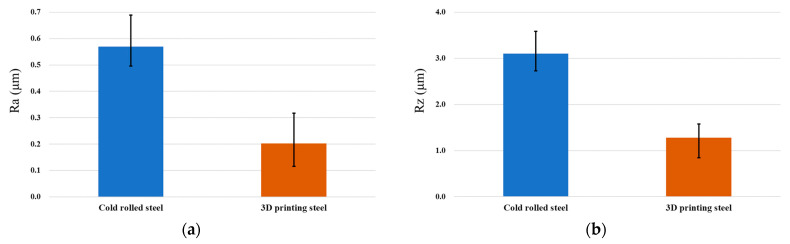
The surface roughness in Ra and Rz after grinding of the surface of the block specimens. (**a**) The surface roughness in Ra. (**b**) The surface roughness in Rz.

**Figure 10 micromachines-15-00353-f010:**
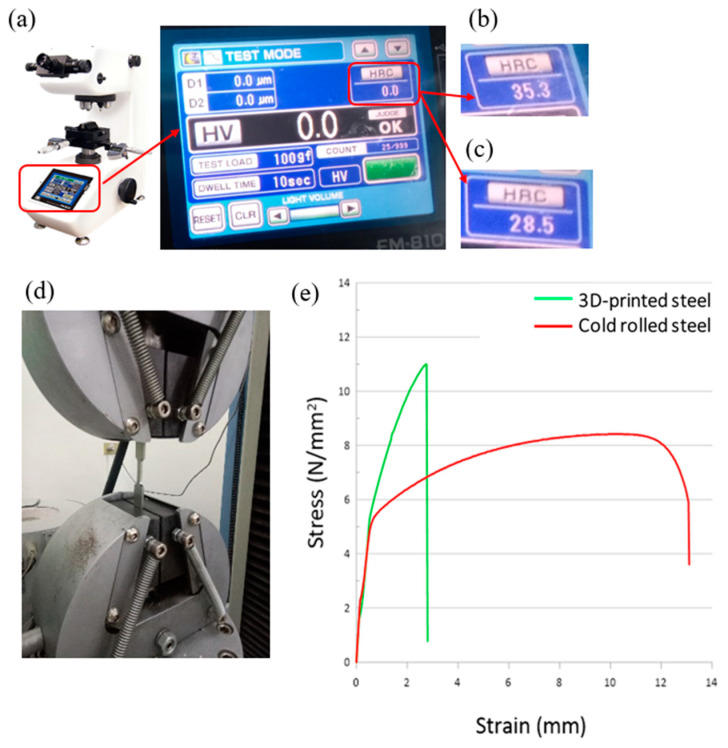
(**a**) is FM-810e Microhardness tester. (**b**) shows the hardness of 3D-printed steel made of 316L SS. (**c**) shows the hardness of rolled steel made of 316L SS. (**d**) is the tensile test. (**e**) is the relationship between stress and strain for the tensile specimens of 3D-printed steel and rolled steel made of 316L SS.

**Figure 11 micromachines-15-00353-f011:**
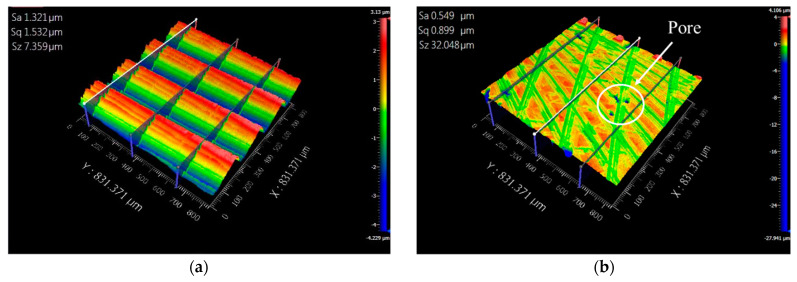
The surface morphology of the block specimen when the block specimen is manufactured by 3D additive printing and roll process after the milling machine cutting of the surface of the block specimen. (**a**). The surface morphology of the block specimen when the block specimen is manufactured by 3D additive printing with 316L SS and milling machine cutting of the surface of the block specimen. (**b**). The surface morphology of the block specimen when the block specimen is manufactured by roll process with 316L SS and milling machine cutting of the surface of the block specimen.

**Figure 12 micromachines-15-00353-f012:**
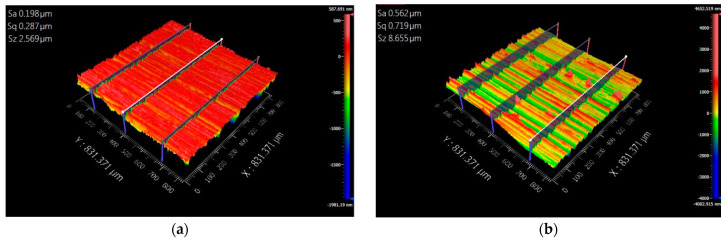
The surface morphology of the block specimen when the block specimen is manufactured by 3D additive printing and rolled after machine grinding of the surface of the block specimen. (**a**). The surface morphology of the block specimen when the block specimen is manufactured by 3D additive printing with 316L SS and machine grinding of the surface of the block specimen. (**b**). The surface morphology of the block specimen when the block specimen is manufactured by rolling process with 316L SS and machine grinding of the surface of the block specimen.

**Table 1 micromachines-15-00353-t001:** Processing parameters of specimen by SLM manufacturing.

Printed Area	Laser Power Watt	Scanning Speed mm/s	Laser Diameter µm
Border	100	250	0.1
Hatches	100	250	0.1
In skin			
Blocked path	220	900	0.1
Border	220	900	50
Additional border	220	900	50
Fill contour	220	900	50
Hatches	220	900	50
Down skin			
Blocked path	100	900	50
Border	220	900	50
Additional border	100	900	50
Hatches	220	900	50
First layer			
Blocked path	100	250	1
Border	100	250	1
Additional border	100	250	1
Fill contour	100	250	1

**Table 2 micromachines-15-00353-t002:** Chemical compositions of 316L stainless steel (wt%).

Material	Fe	Mo	Ni	Mn	Cr	Si	O_2_	C	P	S	N
Cold rolled steel	Bal.	2.1	10.12	1.6	16.74	0.58	-	0.014	0.037	0.002	0.021
3D-printed steel	Bal.	2.5	12.7	1.4	16.8	0.7	0.06	0.01	-	-	-

**Table 3 micromachines-15-00353-t003:** The specimen process conditions by milling and grinding processing using 316L SS.

Milling processing	Milling cutting tool Ø12 × Spindle speed 2500 rpm
Grinding processing	CBN 325N 100B Grinding wheel Ø180 × Rotating speed 2500 rpm

**Table 4 micromachines-15-00353-t004:** Comparison of the performance of the surface roughness in Ra/Rz with 316L SS.

Manufacture Process	Rz/Ra before Milling or Grinding	Rz/Ra after Milling	Rz/Ra after Grinding
3D additive steel	7.08~6.69	4.97	6.35
Cold rolled steel	-	3.38	5.46

## Data Availability

The authors declare that the data sources come from design, simulation, and experiments. We have not plagiarized others.

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
