# Peer review of "Comparing the Performance of Rolled Steel and 3D-Printed 316L Stainless Steel"

_micromachines, 2024, doi:10.3390/mi15030353_

Round 1

Reviewer 1 Report (Previous Reviewer 3)

Comments and Suggestions for Authors

The authors of the article worked very well on the text. I no longer have questions and comments.

Author Response

Response: Thank you for your approval.

Reviewer 2 Report (Previous Reviewer 5)

Comments and Suggestions for Authors

Dear Authors!

My recommendations:

1) instead of "general rolled steel" use the terms "as rolled steel" and instead of "non-traditional" additive manufacturing process use "conventional" manufacturing process

2) in order to compare the mechanical properties (hardness, tensiles, ductility, maleability etc.) of the material specimens should be clarified the dimensions of the rolled material (raw material), the cross-section reduction after roling, the rolling technology (cold or hot), the resulting grain size etc., as the rolling technology also affects these properties.

Comments on the Quality of English Language

Technical language needs improvement

Author Response

Reviewer_02 comments and responses

2-1

Instead of "general rolled steel" use the terms "as rolled steel" and instead of "non-traditional" additive manufacturing process use "conventional" manufacturing process

Response: Thank you for your lovely comments. We have revised these issues as yellow highlights of the words in the resubmitted manuscript.

2-2

In order to compare the mechanical properties (hardness, tensiles, ductility, maleability etc.) of the material specimens should be clarified the dimensions of the rolled material (raw material), the cross-section reduction after roling, the rolling technology (cold or hot), the resulting grain size etc., as the rolling technology also affects these properties.

Response: Thank you for your lovely comments. The dimensions of raw material (316L SS) are W350mm*L1,219mm*t3.0mm in tensile specimens. The dimensions of impact specimens are W1,219*L3,048mm*t10.0mm in 316L SS. The rolling technology was used in the cold rolled process.

The cross-section reduction is about 15-25% after the rolling process.

The grain size is about 20-50µm.

We have revised these issues as yellow highlights of the words in the resubmitted manuscript.

2-3

Comments on the Quality of English Language.

Response: Thank you for your lovely comments. We have revised the quality of the English language. The quality of the manuscript is also collected by the native editor, who uses artificial intelligence software to double-check.

Round 2

Reviewer 2 Report (Previous Reviewer 5)

Comments and Suggestions for Authors

Nice job.  Congratulation. Further success.

This manuscript is a resubmission of an earlier submission. The following is a list of the peer review reports and author responses from that submission.

Round 1

Reviewer 1 Report

Comments and Suggestions for Authors

The examination of 3D printed 316L stainless steel in this study appears overly superficial, resembling more of a technical report. Regrettably, I cannot endorse its publication.

Reviewer 2 Report

Comments and Suggestions for Authors

This is a generally well-written paper and subject to changes suggested below, it can be published.

1.     First sentence is oddly-phrased and can be deleted.

2.     Title can be revised to “Comparing Performance of rolled steel and 3D printed 316L Stainless Steel”.

Comments on the Quality of English Language

The English writing needs to be improved.  Start with first sentence of the abstract that is phrased strangely.

Reviewer 3 Report

Comments and Suggestions for Authors

The manuscript discusses the challenges of Powder Bed Fusion (PBF) with metal powders in 3D additive manufacturing, particularly the surface roughness effect caused by printing parameters. The study compares the qualities of warpage, weight, and surface roughness after milling and grinding processes under the same materials between rolled steel and 3D additive steel. The research shows that 3D additive manufactured parts are approximately 13% to 14% lighter than rolled steel, and the surface roughness performance is better than rolled steel after milling or grinding processing. The study concludes that 3D additive manufacturing can use surface processing to optimize surface performance and achieve a lighter function.

The relevance of the article lies in the growing interest in 3D additive manufacturing and its potential applications in various industries. The novelty of the article is in its comparison of qualities between rolled steel and 3D additive steel, providing insights into improving the surface roughness issue of 3D metal printing.

There are some comments to the text of the manuscript.

1.      Please more clearly formulate the purpose of the work at the end of the Introduction section.

2.      Figure 6 seems redundant (what is stated in the text is sufficient: “…are lighter 13.83%...”). The same applies to Figures 7 and 8 - it is enough to describe the data obtained in the table. These drawings look too school-like.

3.      Paragraph 4. You describe the experimental methods in the results section. It would be more appropriate to present experimental methods at the beginning of paragraph 3, in some section “Materials and methods”.

4.      The conclusions look too stingy and poor.

5.      In your conclusions, you mentioned laser power modes and scan speeds as tools for solving the porosity issue. But there was nothing about this in the text of the article. It would be appropriate to develop these theses in the Discussion of Results section and briefly repeat them in the conclusion.

6.      Overall, your text could have been better structured. The typical structure of a text is Introduction, Materials and Methods, Results, Discussion and Conclusion. In the Introduction section, an overview of the state of research is given and the purpose of the work is formulated (it was not clearly stated by you). The Materials and Methods section describes your research methods (currently they are “spread out” throughout the text). It would also be necessary to pay attention to the English language and the overall style of the text. You can use artificial intelligence to improve your writing style. This will make your (admittedly interesting) work more understandable to readers.

Reviewer 4 Report

Comments and Suggestions for Authors

Dear Authors.

Congratulations on your work, which I found interesting.

Congratulations on your work. Manuscript: Research on the Performance within Processing and Post-pro-2 cessing for 3D printed 316L Stainless Steel; it is well written with an adequate structure as a scientific paper demands.

I have some minor revisions to propose to you to improve your work. Please refer to the following comments:

At work, I lack the results of strength tests that would confirm the application possibilities of structural elements produced by 3D printing and the validity of further research on printing parameters to eliminate or minimize the occurrence of pores in the manufactured elements.

Cross-sectional images of the produced samples should also be taken to determine the number and size of pores.

I also need more information regarding production costs and the economic feasibility of using additive methods for the indicated applications, including post-processing.

Additionally, there needs to be a dimension in Figure 1c - the thickness of both samples.

Reviewer 5 Report

Comments and Suggestions for Authors

Dear Authors, my comments are:

1. I miss the origin and the description of the raw materials:

-powder manufacturer and powder dimensions (diameter)

-rolled AISI 316L steel bars (specimens) manufacturer, dimensions, delivery conditions, heat treatment conditions, microstructures

2. How and whit what equipement did you carried out the analysis of the chemical composition presented on Table 1?

3. How was determined the weight described in Figure 6 ?

4. What type of machines was used for milling and grinding of the specimens?

5. What is the innovation and novelty of the paper? Should be explained in more detail .

6.The conclusion section is too short, must be improvee, should be enriched with the results obtained, give the innovation and novelty of the research described in the manuscript.

7. I recommend reediting the manuscript according to the following chapters:

Introduction

Materials and methods

Results and discussion

Conclusions